# Selective hydrogenation via precise hydrogen bond interactions on catalytic scaffolds

Song Shi[1,2,4], Piaoping Yang[1,4], Chaochao Dun [3], Weiqing Zheng[1], Jeffrey J. Urban [3] & Dionisios G. Vlachos [1] ✉

The active site environment in enzymes has been known to affect catalyst performance through weak interactions with a substrate, but precise synthetic control of enzyme inspired heterogeneous catalysts remains challenging. Here, we synthesize hyper-crosslinked porous polymer (HCPs) with solely -OH or -CH$_3$ groups on the polymer scaffold to tune the environment of active sites. Reaction rate measurements, spectroscopic techniques, along with DFT calculations show that HCP-OH catalysts enhance the hydrogenation rate of H-acceptor substrates containing carbonyl groups whereas hydrophobic HCP-CH$_3$ ones promote non-H bond substrate activation. The functional groups go beyond enhancing substrate adsorption to partially activate the C = O bond and tune the catalytic sites. They also expose selectivity control in the hydrogenation of multifunctional substrates through preferential substrate functional group adsorption. The proposed synthetic strategy opens a new class of porous polymers for selective catalysis.

Enzymes-inspired catalysts composed of active sites and protein binding pockets interacting with a substrate have been a long-standing goal of heterogeneous catalysis (Fig. 1a)[1–3]. Such artificial analogs require the synthesis of spatial structures with suitable electronic properties[4,5]. There has been increasing emphasis on tuning the environment of active sites at the nanoscale[5–7] for instance, metal−organic frameworks (MOFs) with tailorable coordination building blocks[8,9]. For example, UiO-66 modified with poly(dimethylsiloxane) (PDMS) becomes hydrophobic and enhances the concentration of certain organic substrates and the activity[10]. The accurate control of reactions by engineering the active site environment has recently been reviewed[8]. Engineering single-atom catalysts through the local coordination and electronic state of the catalytic center is another class of tunable heterogeneous catalysts[11]. Beyond geometric structure, it is even more desirable to mimic the promotion mechanism of enzymes by tuning the weak interactions between the reactant and the environment[12]. Hydrogen bond plays a key role in enzyme catalysis

especially for low-barrier hydrogenations[13]. For example, the amino acid residue in the human transketolase at position 366 could form a crucial hydrogen bond with the N1 in the substrate thiamine (Fig. 1c)[14]. Due to the lack of structural precision in synthesizing traditional solid catalysts, mimicking the weak interaction to modulate catalytic reactions remains challenging. The precise control of the active site environment requires unambiguous binding motifs to enable the incorporation of catalytic active sites, well-adjustable chemical composition with specific functional groups, high surface area with hierarchical porosity for fast mass transfer, and a stable catalyst during the reaction and recycling. Porous organic polymers (POPs) are multi-dimensional porous network materials built via strong covalent linkages between various organic building blocks. They have recently emerged as versatile, tailorable materials[15] with unique 3D porous structure, by changing the monomer or the functional groups. For example, by encapsulating metal nanocrystals in amine-based POPs, Cargnello's group found a transition in the Pd-catalyzed CO

[1]Department of Chemical and Biomolecular Engineering and Catalysis Center for Energy Innovation (CCEI), University of Delaware, Newark, DE 19716, USA. [2]State Key Laboratory of Catalysis, Dalian Institute of Chemical Physics, Chinese Academy of Sciences, Dalian 116023, People's Republic of China. [3]The Molecular Foundry, Lawrence Berkeley National Laboratory, Berkeley, CA 94720, USA. [4]These authors contributed equally: Song Shi, Piaoping Yang. ✉e-mail: vlachos@udel.edu

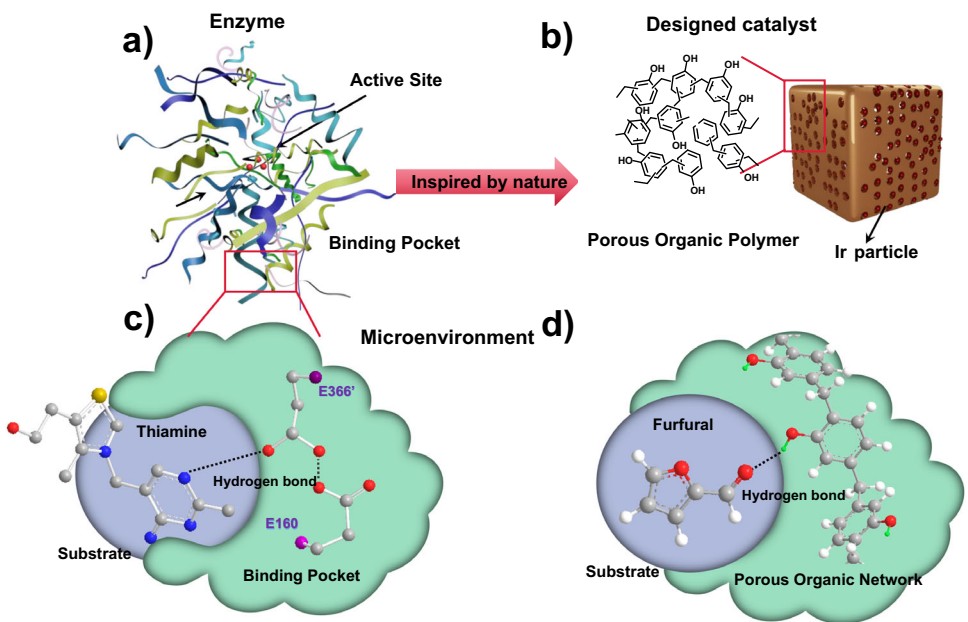

**Fig. 1 | Enzymes inspired microenvironment-controlled catalyst.** Schematic diagram of (**a**) enzyme structure and (**b**) designed catalyst porous catalyst. **c** Hydrogen bond interaction between the substrate thiamine and the human transketolase. **d** Hydrogen bond interaction between furfural and the porous organic polymer.

oxidation[16,17]. Xiao and Ma synthesized POPs with building blocks containing the same functional groups as the high boiling solvent used in the fructose dehydration[18]. Their approach provided a "solid solvent" like character to the active site, greatly promoting fructose dehydrogenation while avoiding a high boiling solvent. All these examples demonstrate the ability of POPs to be good models of enzyme-inspired catalysts.

Herein, we aim at designing active sites with precisely controlled hydrophobic or hydrophilic environments to form a hydrogen bond with substrates that act as H acceptors (Fig. 1b, d). We synthesize hyper-crosslinked porous polymer (HCPs) with nitrogen (N) atoms for anchoring metal nanoparticles and methyl ($CH_3$) or hydroxyl (OH) groups to direct the chemistry. Through spectroscopic techniques, reactions of various substrates, and density functional theory (DFT) calculations, we demonstrate a substrate-specific H bond that not only affects the substrate adsorption but also regulates the hydrogenation rate and selectivity. Our synthetic strategy opens the possibility of precise tuning catalytic scaffolds for improving hydrogenations.

## Results

### Tuning active site environment and characterization

HCPs with -OH functional groups (HCPs-OH) were synthesized through the Friedel-Crafts alkylation using phenol as the monomer. A theoretical ratio of 20% triphenylamine was used as part of the monomer enabling the N site to act as the binding site for metal nanoparticles. As a control catalyst, HCPs with $CH_3$ groups (HCP-$CH_3$) were synthesized with the same method using toluene monomer. The structural integrity of HCPs and their functional groups were demonstrated via solid-state (ss) $^{13}C$ Nuclear Magnetic Resonance (CP/MAS $^{13}C$-NMR, Fig. 2a) and Fourier-transform infrared spectroscopy (FT-IR, Fig. 2b). Both of the two catalysts show a wide resonance peak near 135 ppm in Fig. 2a, which is attributed to the aromatic carbons and the one at 36 ppm to the methylene carbon formed by the Friedel-Crafts reaction. The main absorption peaks (1450, 1500, and 1600 $cm^{-1}$) in the FT-IR spectra in Fig. 2b originate from the aromatic ring skeleton vibrations, and peaks (2926, 2850, and 1472 $cm^{-1}$) are related to the asymmetrical and symmetrical stretch vibrations and deformation vibration of the methylene group. These are consistent with the NMR results, illustrating they

have similar skeletons. For HCP-OH, the peak at 150 ppm is ascribed to the carbon connected to -OH groups, which is also reflected in FT-IR by the wide peaks at 3500 $cm^{-1}$ are ascribed to the -OH groups and the peaks near 1210 $cm^{-1}$ correspond to the C-O stretching band of the phenolic hydroxyl. For HCP-$CH_3$, the peak at around 18 ppm is due to the methyl carbon connected with the benzene ring, also shown in the FT-IR peaks at 2980 $cm^{-1}$. Note no signal is located at 160-200 ppm, suggesting no carboxylic acid or aldehyde groups.

Taken together, the ss-NMR and FT-IR results indicate the synthesis of an HCP scaffold with desirable functional groups. The functional groups are also manifested in simple wettability tests (Fig. S1). HCP-$CH_3$ floats on water and has a water contact angle of 107°. In contrast, HCP-OH is hydrophilic with a water contact angle of 33° and disperses homogeneously in water.

Fig. 2c reveals the rough surface of HCPs-OH with interconnected pores (SEM images) and the 3D reconstruction using FIB (Fig. 2d), clearly showing their hierarchical structure to enable fast transport. O and C are homogeneously distributed, i.e., the OH groups are evenly distributed (Fig. 2f, h). The porous properties were confirmed via $N_2$ physical adsorption isotherms (Fig. S2). The steep increase uptakes at low relative pressure ($P/P_0 < 0.001$) indicate abundant microporosity, as expected. The corresponding BET-specific surface area, total pore volume, the average pore width, and the element ratios (C, H, and N) are listed in Table S1. The existence of N illustrates the incorporation of triphenylamine as binding sites and the similar fraction of N (~0.7 wt%) in the HCPs of different functional groups demonstrates the same number of binding sites. Ir nanoparticles were introduced via the impregnation and reduction method (Fig. 1; Ir-HCP-OH and Ir-HCP-$CH_3$) and were homogeneously distributed inside the HCPs (TEM images in Fig. 2g and Fig. S3, and no obvious peaks of Ir in XRD patterns in Fig. S4 also confirmed Ir was in its high dispersed state). A slight BET area decrease was observed upon loading Ir but the porous structure was preserved (Table S1, Fig. S2). Overall, two catalyst supports with OH or $CH_3$ groups were successfully synthesized.

### Selective hydrogenation

The TGA curves show that both catalysts are stable below 300 °C (Fig. S5), i.e., they are stable under our reaction conditions. We

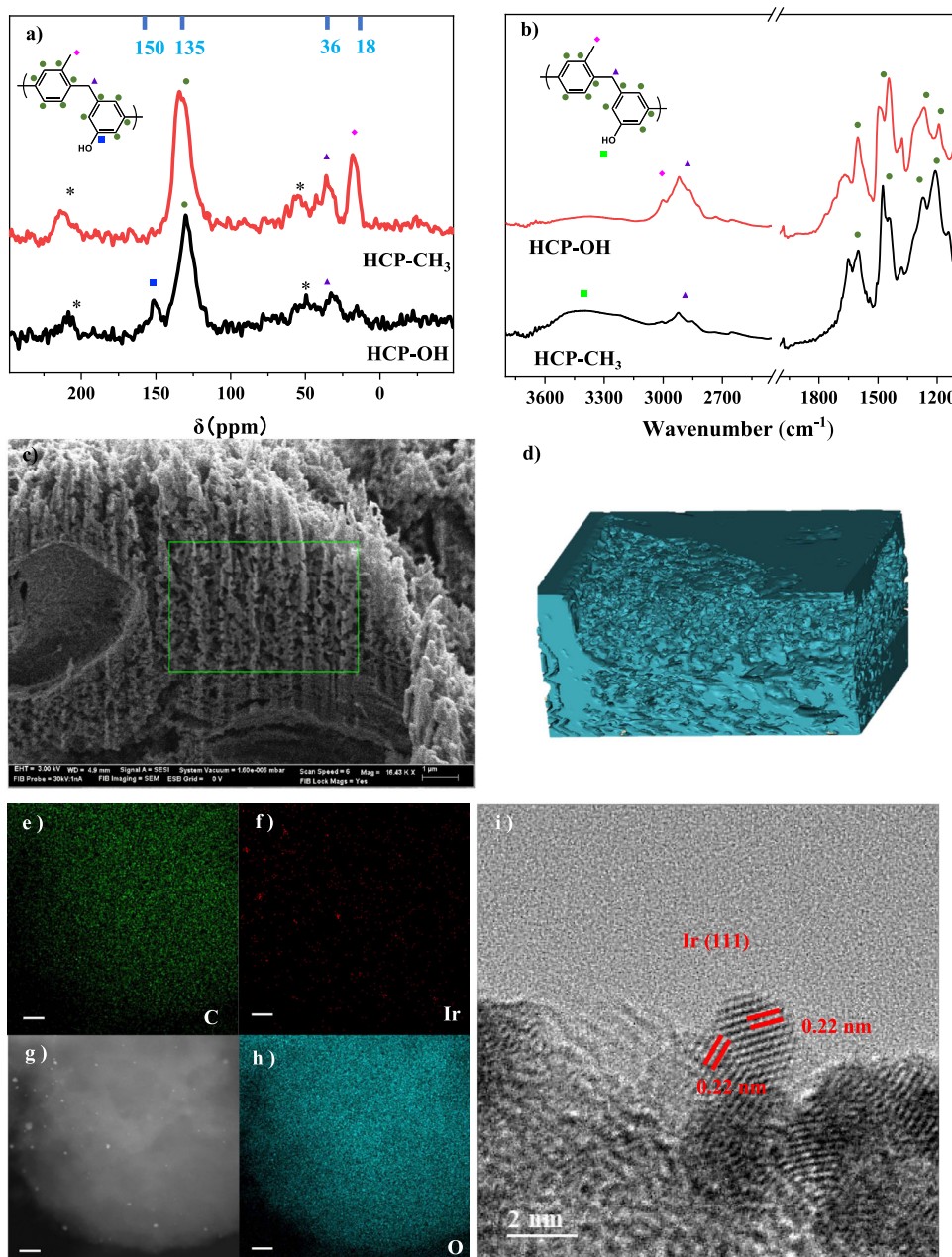

**Fig. 2 | Catalyst structure and composition. a** Solid-state $^{13}C$ CP/MAS NMR of HCP-OH and HCP-CH$_3$. Asterisks denote spinning sidebands. **b** FT-IR of the HCP-OH and HCP-CH$_3$. **c** SEM images of HCP-OH. The green square is the FIB reconstruction area. **d** 3D reconstruction through SEM-FIB. **e–h** HADDF and TEM mapping of different elements of Ir-HCP-OH. **i** HR-TEM images of Ir particles.

employ different substrates to understand the impact of functional groups on activity. Toluene hydrogenation over Ir-HCP-CH$_3$ is faster than Ir-HCP-OH (Fig. 3a), consistent with prior work[10] that hydrophobicity enhances the activation of weak polar hydrocarbon substrates. In sharp contrast, furfural hydrogenation shows an inverse promotion effect (Fig. 3b), and 2-methyl furan (2-MF) ring hydrogenation exhibits similar activity (Fig. S6). The strong substrate-specificity over the Ir-based catalysts (Fig. 3d) underscores the impact of the active site environment. For carbonyl-containing substrates, the hydrophilic support shows a superior reaction rate. For less polar substrates, like toluene, a hydrophobic catalyst enhances the rate.

To eliminate the potential effect of the Ir active site, the detailed particle size and the electronic density of Ir were further studied. We believe that the slightly different average particle size (around 3 nm;

Fig. S3) does not dramatically affect selective hydrogenation. XPS data (Fig. S8; no significant shift in the binding energy at 60.5 eV, ascribed to the Ir$^0$ nanoparticles)[19] and CO adsorption Drifts-Ir data (Fig. S9, both of the two catalysts show adsorption peaks at 2044 cm$^{-1}$) illustrated a similar electronic density. Overall, Ir-HCP supports with similar active metal sites and tuneable microenvironments (hydrophilic vs. hydrophobic) were built.

Although the two catalysts are similar in particle size and electronic state, other aspects of the active site, like the crystal phase, the spatial location, and so on, could affect the reaction. It is not possible to rule them out. We tested Pd and Pt nanoparticles for furfural and toluene hydrogenation (Fig. 3c and corresponding characterizations shown in Figures S10 and S11). Although the absolute enhancement is metal specific, as expected, the promotion effect is still seen: the HCP-OH catalysts enhance the furfural hydrogenation rate, whereas the

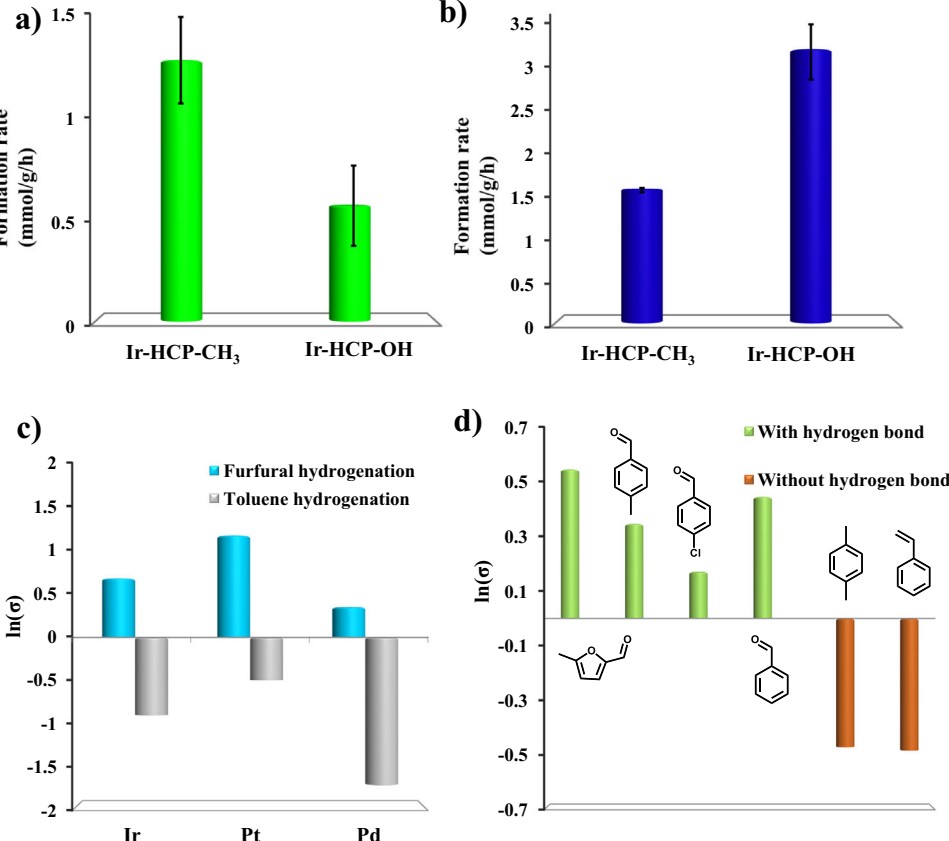

**Fig. 3 | Catalyst performance for different substrates.** Reaction rate difference (**a**) in toluene hydrogenation, (**b**) in furfural hydrogenation, (**c**) over different metals, and (**d**) of multiple aromatic substrates. σ = Rate$_{Ir-HCP-OH}$/Rate$_{Ir-HCP-CH3}$,

Reaction conditions: substrate, 1 mmol; catalyst, 40 mg; hexane, 10 mL; H$_2$, 300 psi; 120 °C. In panels **a**, **b**, and **d**, the catalyst is Ir. The reaction is shown in Fig S29. Error bars are standard errors (SE) obtained from repeated experiments.

HCP-CH$_3$ ones the toluene hydrogenation rate. This finding showcases that the main driver for the reaction rate enhancement stems from the environment of the active site and is independent of the metal. All of this evidence showed that the active site is not the main reason for the reaction difference.

To assess the impact of active site environment on adsorption, the isotherms of furfural and toluene were obtained in the reaction solvent (Fig. S12, S13). The results are revealing: furfural adsorbs on HCP-OH catalyst twice (a saturation adsorption of 1.95 mmol/g) than on HCP-CH$_3$. The affinity constant K of HCP-OH is also about 3x than that of HCP-CH$_3$. In contrast, the toluene saturation on the two supports is similar and lower than furfural's (Fig. S13), illustrating the group-specific interaction of furfural with HCP-OH.

To further understand the substrate-active site interaction, in situ DRIFTS furfural adsorption was conducted (Fig. 4a, b). Both supports show furfural adsorption, but the peaks over HCP-OH are much higher, consistent with the adsorption isotherm (Fig. S14). Those around 1700 cm$^{-1}$ were ascribed to the carbonyl group of furfural[20]. The multiple peaks are characteristic of multiple binding modes. Deconvolution shows that the one at 1720 cm$^{-1}$ is due to the physically adsorbed furfural, those at 1698 cm$^{-1}$ to the furan ring adsorption, and the lowest one at 1670 cm$^{-1}$ to the O terminated C = O group (η$^1$)[20,21]. No peaks at 1450 cm$^{-1}$, characteristic of η$^2$ adsorption (with O and C of the carbonyl group adsorbed), were observed. It is noteworthy that all peaks on the HCP-OH support shift to a lower wavenumber than HCP-CH$_3$, demonstrating stronger furfural adsorption (Fig. S14). In addition, the ratios of the three adsorption modes on the two catalysts differ (Fig. 4a, b). Specifically, the η1 conformation on HCP-OH is higher than on the HCP-CH$_3$, i.e., the interaction of the C = O group is stronger (or there are more binding sites). Furthermore, with increasing

temperature, all the furfural peaks on the HCP-CH$_3$ disappear at 130-170 °C (Fig. S15), whereas on the HCP-OH, the desorption temperature increases to 170-220 °C (Fig. S16). The furan ring-related peaks are similar on HCP-OH and HCP-CH$_3$, i.e., the binding differences are mainly due to the C = O group (Fig. S14). ATR-IR adsorption (Fig. S17) shows that the furfural peaks on the HCP-CH$_3$ are like those of the pure liquid furfural. On the HCP-OH, the ring peaks are also pure furfural like, but the C = O peaks at 1670 cm$^{-1}$ shift to a lower wavenumber and become broader, i.e., the C = O group binds stronger due to different binding sites. Since the main difference between supports lies in their functional groups, the adsorption difference is due to the carbonyl group interacting with the substrate via H bond. Due to the precise composition of the scaffold enabled by tuning the monomer, one can mimic heterogeneous surface functional groups. If a hydrogen bond is implicated, the most powerful evidence can come from concentration-dependent $^1$H-NMR[22]. As shown in Figs. 4c and 4e, the active H of phenol is located at 4.56 ppm. With the addition of furfural, the H peaks shift to the low field of 5.42 ppm, undisputedly illustrating H bond formation. A ΔG of −0.552 kcal/mol (Fig. S18) was calculated from $^1$H-NMR shift (Eqs. S2 and S3). On the contrary, the active H did not shift when toluene was added to the solution, indicating no H bond of toluene with OH. As a control experiment, 2-MF gives a tiny shift in the H peak (△G = −0.213 kcal/mol, Fig. 4d, and S19), suggesting a relatively weak H bond with phenol (Fig. 4d). Therefore, the H bond strength difference between the OH bond and the substrate is the main reason for the reaction difference, which was different from the previous modulation method (Table S4).

Density Functional Theory (DFT) calculations elucidate the effect of OH and CH$_3$ groups on the adsorption of furfural. Since POPs are amorphous and hard to model computationally, we focus

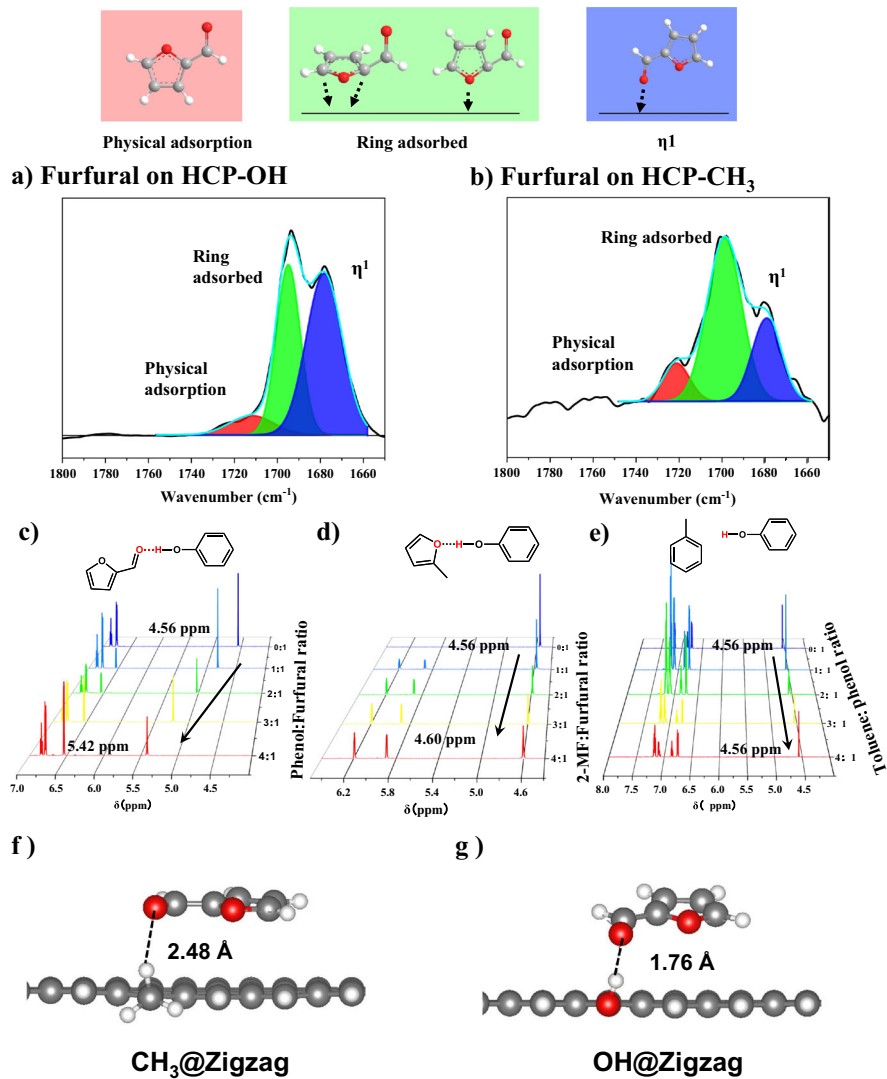

**Fig. 4 | Hydrogen bond interaction of the substrate with different scaffolds.** Drifts IR of furfural on (**a**) HCP-OH and (**b**) HCP-CH₃. Concentration-dependent ¹H-NMR results of (**c**) furfural with phenol, (**d**) 2-MF with phenol, and (**e**) toluene with phenol. Optimized configurations of furfural adsorption over (**f**) CH₃ group and (**g**) OH group on zigzag graphene models. The cartoons at the top indicate the adsorption modes.

on the functional groups of POPs rather than the detailed spatial configuration. Therefore, the 2D graphene ribbons with zigzag and armchair edges and one terminal H atom being replaced with an OH or a CH₃ group were employed as computational models to capture the functional groups of HCP-OH and HCP-CH₃ support, respectively (Fig. S20), considering that they not only capture the interaction between furfural and the two functional groups but also contain aromatic rings found in POPs. The optimized configurations of furfural adsorption illustrate that over the CH₃ group, the furfural is parallel to the graphene surface (Fig. 4f and S21a) with a flat-lying geometry, suggesting rin g adsorption mode. Over the OH group, however, furfural slightly tilts with the carbonyl group towards the OH group, indicating η1 adsorption mode (Fig. 4g and S21b). These agree qualitatively with the adsorption isotherm, TPD, and DRIFTS-IR results above. Additionally, the adsorption energies of furfural on OH groups are lower than on CH₃ groups by 0.07 eV for the zigzag model and 0.16 eV for the armchair model (Table S3). The distance of the OH group from the carbonyl O atom of furfural is 1.76 Å on the zigzag model and 1.85 Å on the armchair model, shorter than the distance between the CH₃ group and the carbonyl O atom of furfural (Fig. 4f-g and Fig. S18a-b), suggesting a stronger interaction between furfural and OH group due to the hydrogen bonding of

$-OH\cdots O = CH -$ . The hydrogen bond interaction also elongates the bond length of C = O from 1.230 Å to 1.242 Å, weakening the C = O bond. These results that OH group can strengthen the binding of furfural by hydrogen bond and weaken the C = O bond were further verified by studying the interaction between furfural and the two monomers (toluene and phenol) (Fig. S22). This clearly shows that the OH groups play a role beyond adsorption in catalytically promoting the C = O bond hydrogenation. We return to this point below.

**Importance of OH density**
The above results clearly show that the functional groups go beyond affecting adsorption. To gain further insight, the -OH density was varied by adjusting the ratio of phenol to toluene in the synthesis. The OH density difference was confirmed using ssNMR. The ratio of the peaks at 150 ppm (benzene ring carbon connected with OH) and 135 ppm (total benzene ring) reflects the OH density[23]. The peak ratio increases with increasing phenol content and so is the furfural hydrogenation rate (Figs. 5a,5b). These results underscore the importance of the density of the OH on catalytic performance and the tunability of the active site environment.

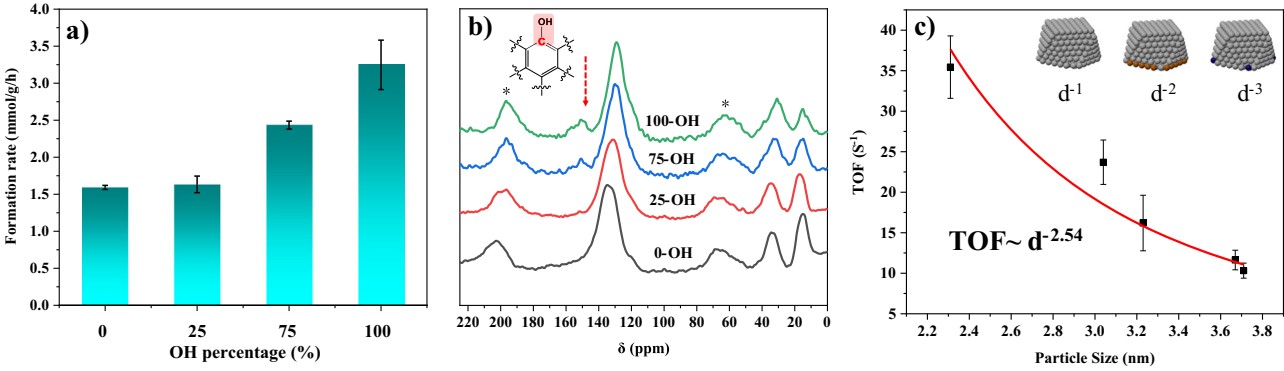

**Fig. 5 | Effect of OH on reaction performance. a** Formation rate of furfuryl alcohol vs. OH density. **b** Solid-state $^{13}$C CP/MAS NMR of the HCP with varying OH density. **c** TOF vs. particle size. Error bars are SE obtained from repeated experiments.

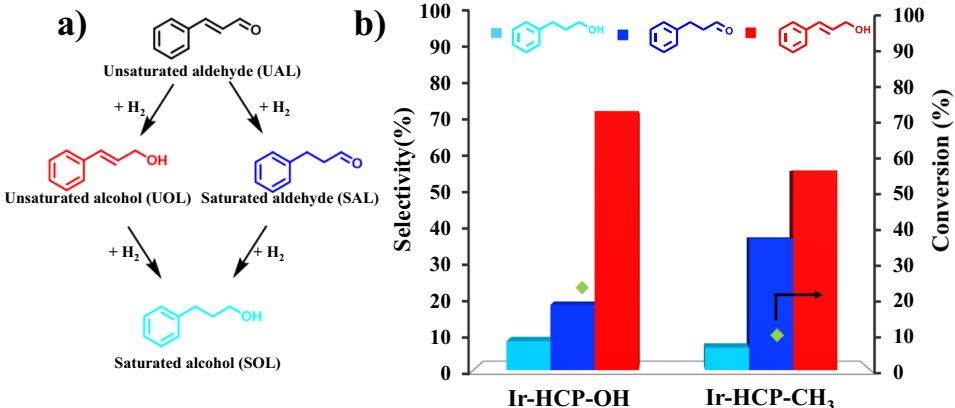

**Fig. 6 | Hydrogen bond effect on selectivity. a** Reaction routes of cinnamaldehyde hydrogenation. **b** Selectivity difference over Ir-HCP-OH and Ir-HCP-CH$_3$ in the cinnamaldehyde hydrogenation.

## Nano-scaled intimacy of OH groups to active sites

To further understand the function of the OH group on the reaction, the particle size effect of Ir-HCP-OH was investigated. The number of exposed surface, edge, and vertices (corner) sites to the total number of atoms is proportional to ~$d^{-1}$, $d^{-2}$, and $d^{-3}$, respectively[24,25]. By controlling the loading of Ir, five catalysts of different particle sizes were synthesized (Fig. S23). The results show that TOF scales as $d^{-2.54}$ (Fig. 5c), indicating potentially that Ir edge sites near OHs are more active, clearly showing a synergistic effect between the two sites, consistent with the DFT results above.

## Selectivity control

Aside from the rate control, the hydrogen bond interaction can also be leveraged in selectivity control. In the α,β-unsaturated aldehyde hydrogenation, it is a challenge to selectively hydrogenate the C = O over the C = C bond[26]. Conformation control provides an efficient method to promote the C = O hydrogenation selectivity. Previously, self-assembled monolayers (SAMs) were used for cinnamaldehyde hydrogenation[27]. With the phenylated SAMs, the π-π interaction between SAMs and the substrate promotes the "head down" conformation, resulting in higher selectivity[28]. Here, we exploit the hydrogen bond between OH and C = O groups (Fig. 6a). Not only does the hydrophilic catalyst increase the hydrogenation rate, but it also promotes the selectivity to the unsaturated alcohol (Fig. 6b). This result shows hydrogen bonds could regulate selective hydrogenation of multifunctional substrates.

## Catalyst Recyclability

Finally, the recyclability of Ir-HCPs-OH was tested. The after-reaction catalyst TEM (Fig. S24) showed some big particles forming. The selective hydrogenation of furfural was tested at a low conversion of about 15% (Fig. S25). The catalytic activity remains constant in run 2, but the conversion drops in run 3, due to the structure collapse or sintering of the Ir nanoparticles. Hot filter experiments and ICP show that the catalysis is heterogeneous with no detectable leaching (Fig. S26). Catalyst regeneration from sintering can be overcome by redispersion or encapsulation methods developed recently[29–31].

## Discussion

In enzyme catalysis, the weak interaction of the substrates with the pocket is essential in performance. Controlling the spatial interactions of substrates with the surface functional groups of the scaffold catalysts with precision is crucial. In many traditional catalysts, such as activated carbon, there are many functional groups[32]. Although one could determine their ratio, it is hard to distinguish the role of each. In this regard, POPs are an ideal platform for tuning the environment around the active site. The H-bond interaction between the substrate and the catalyst is not limited to POPs; rather, we have found this H bond also in zeolite materials (Supply note 1), illustrating universality. We tested the adsorption of ketones (cyclopentanone, cyclohexanone, and cycloheptanone) on Si-Beta and Si-ZSM-5 (Si/Al = 500) using in-situ FT-IR in Fig. S27. Irrespective of the substrate, Si-Beta adsorbs almost 10× than Si-ZSM-5 (Table S2) despite minimal differences in BET surface area, pore volume, and pore size (Table S1). DRIFTS-IR reveals a

huge difference in the area around 3500 cm$^{-1}$ (Fig. S28), ascribed to Si-OH groups, implying many more OH groups in Si-Beta than Si-ZSM-5 that can hydrogen bond with the carbonyl groups.

In summary, inspired by enzyme catalysis, we designed HCPs-based catalysts with different environments consisting of precisely grafted OH and CH$_3$ groups around the active sites whose density can change. These catalysts expose substrate-specific catalytic performance. Hydrophilic catalysts form H-bonds with C=O containing substrates like furfural, absent from weakly polar substrates. Such interactions enhance adsorption, potentially modify the interfacial sites to increase reaction rates, and promote selective C=O hydrogenation over C=C hydrogenation in α,β unsaturated and furanic aldehydes. This work provides a new strategy for bio-inspired catalysts, by taking advantage of the weak interactions between substrates and the active site environment to direct the chemistry.

## Methods

### Catalyst synthesis

Hyper crosslinked porous polymers (HCPs) were synthesized using the Friedel-Crafts alkylation reaction. Monomer (0.02 mol) was mixed with dimethoxymethane cross-linker (0.04 mol) and dissolved in 1,2-dichloroethane (20 mL). Then, anhydrous FeCl$_3$ (0.04 mol) was quickly added under stirring. The mixture was heated at 45 °C for 5 h and 80 °C for 19 h. After cooling, the solid was washed with methanol for several times and Soxhlet-extracted in methanol for 48 h, and finally dried in an oven at 60 °C for 24 h to dark brown powders.

Ir-HCP-OH and Ir-HCP-CH$_3$ are synthesized with the impregnation method. The procedure is following: a certain amount of H$_2$IrCl$_6$ was dissolved in a mixture of water and ethanol (volume 1:1) and stirred for 30 min, then 500 mg HCPs were added under stirring for 1 h at room temperature. The solution was evaporated and then dried on a 90 °C hotplate. The as-synthesized catalysts were put in a quartz tube and reduced at 553 K for 5 h under a continuous pure H$_2$ flow.

Characterization. X-ray photoelectron spectroscopy (XPS) was performed on a Thermofisher ESCALAB 250Xi spectrometer using AlKα radiation. The binding energies were calibrated using the C 1s level (284.8 eV) as the internal standard reference. $^{13}$C cross-polarization magic-angle spinning nuclear magnetic resonance ($^{13}$C CP/MAS NMR) spectra were collected on a Bruker AVANCE III HD 600 MHz. In situ Drifts spectroscopy measurements were conducted on a 6700 instrument equipped with a Harrick drifts cell. ATR measurements of the catalyst were conducted on a 6700 instrument equipped with a golden state ATR equipment. The TEM images were obtained on the JEM2010F and JEM2100F. The SEM image and FIB images were recorded on Auriga 60. XRD patterns were collected on Bruker D8 with Cu Kα radiation. N$_2$ adsorption isotherms were collected on Micromeritics ASAP 2020 BET Analyzer.

Catalytic reactions were performed in a 120 mL Parr reactor. Typically, 1 mmol of furfural, 20 mL of hexane, and 20 mg of catalyst were added to the reactor. Then, the reactor was charged with 300 psi of H$_2$ and heated to the desired temperature under mechanical agitation. Trimethyl benzenes were added as internal standards. The products were identified by Agilent 7890 N GC/5973 MS detector and quantitated by Agilent 7890 N GC equipped with a CP-Volamine (30.0 m × 0.320μm) and flame ionization detector.

Spin-polarized DFT calculations were performed using the Vienna ab initio software package (VASP, version 5.4.1)[33]. The electron exchange and correlation effects were described by Perdew–Burke–Erzenhof (PBE) exchange-correlation functional[34]. The core electrons were represented with the projector augmented wave (PAW)[35] method and the plane-wave cutoff energy was set at 500 eV. Van der Waals interactions were considered via the dispersion-corrected density functional theory model (DFT-D3)[36]. The Gaussian smearing method with a smearing width of 0.05 eV was employed. The pristine graphene ribbon with armchair edges contained 54 C atoms

and 12 H atoms, while the one with zigzag edges consisted of 60 C atoms and 12 H atoms. OH and CH$_3$ were grafted onto terminal sites of armchair and zigzag graphene ribbons. The vacuum space in the z-direction was set to 25 Å and the in-plane vacuum between ribbons was set to at least 15 Å. The Brillouin zone was sampled with a $(1 \times 3 \times 1)$ k-point and $(3 \times 1 \times 1)$ k-point grid for the armchair model and the zigzag model, respectively. All geometry optimizations were performed using the conjugate gradient algorithm. The atomic force convergence of 0.02 eV/Å and the energy tolerance of 10$^{-6}$ eV were employed. All adsorption energies reported are electronic energies. The energy of gas-phase furfural was calculated in boxes of 20 Å × 21 Å ×22 Å using gamma point. The adsorption energy of furfural ($E_{ads}$) was calculated using

$$E_{ads} = E_{surf + furfural} - (E_{surf} + E_{furfural}) \qquad (1)$$

where $E_{surf+furfural}$, $E_{surf}$, and $E_{furfural}$ are the total energy of the adsorbed furfural and the surface, the energy of the clean surface, and the energy of furfural in the gas phase, respectively.

The interaction strength between molecules were calculated using the Gaussian 09 program[35]. B3LYP/6−311 + G(d,p) was used with basis set superposition error (BSSE) correction.

The interaction energy between furfural and monomer (phenol and toluene) was calculated using

$$E_{inter\_F\_M} = E_{F\_M} + E_{BSSE} - (E_F + E_M) \qquad (2)$$

where $E_{F\_M}$, $E_{BSSE}$, $E_F$, and $E_M$ are the total energy of furfural and monomer (F = furfural, M = phenol or toluene), the BSSE energy, the energy of furfural, and the energy of the monomer, respectively.

## Data availability

All data generated in this study are provided in the a Data file in google drive (https://drive.google.com/drive/folders/12RF1ptZrCMhqsxoelXtCY3BQin5bx9vM).

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

## Acknowledgements

This work was supported as part of the Catalysis Center for Energy Innovation, an Energy Frontier Research Center funded by the US Dept. of Energy, Office of Science, Office of Basic Energy Sciences under award number DE-SC0001004. Dr. Song Shi thanks the support of the "International Talent Program" from the Dalian Institute of Chemical Physics and National Natural Science Foundation of China (Grant no. 22072147).

## Author contributions

S.S. carried out the catalyst preparation, characterizations, analysis, tests and completed the manuscript. P.Y. carried out the DFT calculations. C.D. and J.U. carried out the TEM and EDS mapping. D.G.V., C.D., W.Z. and P.Y. discussed the results and assisted with the manuscript preparation. D.G.V led the project and revised the paper. All authors reviewed and commented on the manuscript.

## Competing interests

The authors declare no competing interests.
