## [Peer Review File · Nature Communications]

Selective hydrogenation via precise hydrogen bond interactions on catalytic scaffoldsREVIEWER COMMENTS

Reviewer #1 (Remarks to the Author):

In this manuscript, Shi and co-workers report a novel catalyst for selective hydrogenation, and tailor the activity and selectivity by augmenting functional groups on the catalyst. Experimental investigations were further appended by DFT calculations. While the idea is novel, I have some observations, issues, and reservations. Some highlights have been provided below:

1. The authors have used the term "enzyme mimetic" on multiple occasions. Is this really appropriate? Enzyme mimetics generally implies that the reaction mechanism is same as that in the enzyme, which has not been attempted to be proved in the present case. See author's own statement: "In summary, inspired by enzyme catalysis, we designed HCPs-based catalysts with different environments consisting of precisely grafted OH and CH₃ groups around the active sites whose density can change." So I would rightly agree that the catalyst is enzyme inspired rather than enzyme mimetic.
2. Page 1: See the correction PMDS instead of PDMS.
3. Figure 2: y-axis title could be better; sigma is not defined. This in fact is a general problem with several figure axes titles.
4. Page 3: "Although the two catalysts are similar in particle size and electronic state...": I highly doubt that it would be even possible to comment upon similarity of their "electronic structures". This statement is not appropriate.
5. Page 5: I would completely disagree with the computational insights provided in the manuscript. There has been no justification on choice of the model and even why graphene was used as a representative of POPs. The DFT analysis is just not appropriate, and for the sake of correctness, I would even be willing to get rid of the computational analysis.
6. Page 5: "This is the first evidence that the OH groups play a role beyond adsorption in catalytically promoting the C=O bond hydrogenation" - This generic claim not correct. There have been several reports which reveal the role of surface functional groups beyond affecting adsorption. See for e.g. Huang et al., Matter, 2019, 1, 1656–1668; Rytter and Holmen, Topics in Catalysis, 2018, 61, 1024–1034; Xu, JACS, 2009, 16366

Reviewer #2 (Remarks to the Author):

The proposed manuscript titled "Selective hydrogenation via precise hydrogen bond interactions on catalytic scaffolds" reports an intriguing work devoted to demonstrating the influence of active site environment on catalytic performance. Mimicking enzyme composed of active sites and protein the authors synthesize hyper-crosslinked porous polymer modified with -OH and -CH₃ groups on the polymer scaffold to tune the interaction between reactant and active sites. This approach is quite interesting. Indeed, a fundamental understanding of the effect on active sites environments on its

interaction with the reactant and on structure-performance relationship will open numerous new opportunities for the development of advanced catalysts. Similar works were reported in the recent years using zeolite and studying the confinement-induced reactivity and the molecular recognition phenomena on these catalysts (see as an overview ChemCatChem 2019,11,134– 156).

The synthesis and characterization of the materials is well reported nevertheless, the following points should be considered by the authors for the publication in a high impact journal as Nature Comm. In particular, the part related to catalytic tests has to be improved since the current presentation leave doubt on the observed effect and on the interpretation and conclusions.

- Preparation method, they used the impregnation method to produce Ir nanoparticles, not to introduce Ir nanoparticles as they write. The used it is not the best method to control the Ir particle size and could cause some variability on the studied materials due to the different hydrophilicity of the polymer support surface where applied. Small difference in nanometal particles dimension or adhesion on support can produce significant difference in catalytic activity.
- The authors should include some schemes with the hydrogenation reaction network and possible products and better discuss in the main text of the objectives for the products they want to target.
- The calculation of conversion, selectivity, TOF and carbon balance should be included. When they use TOF should actually consider the number of surface-active sites calculated by other methods, not the nominal weight of metal. Moreover, possible leaching and experiments with hot filtration to verify that the reaction is not happening due to leached metal. Leached metal can be reabsorbed therefore hot filtration experiment is essential.
- When they discuss catalytic activity, they should discuss in terms of selectivity at iso-conversion or yield, this discussion is missing, so the catalytic data are not well presented.
- Characterisation of the catalysts including Pd and other metals should include except TEM, HRTEM, XPS and with the TEM data should include as well the standard deviation values. This should be for fresh and used catalysts. HAADF analysis is recommended especially for a high-profile paper.
- Have the authors verified that they work are under kinetic regime, e.g. effect of stirring rate, amount of catalyst used and reaction temperature to calculate act. energy? What is the effect of hydrogen pressure and temperature? Have they optimised the reaction conditions? These data should be presented to demonstrate the reliability of discussed comparison.

My main concern is the stability of the catalyst that obviously, for the results the authors present it is very low. If the catalyst is actually not recyclable, the authors should find a way to stabilise it first, otherwise there is no possible to take in account all the discussions about different environment on active site since we are looking to a transient catalytic behaviour.

Moreover, the authors should discuss their work and present their findings with a comparison with the current literature in terms of experimental and computational work, therefore addition of a table. There are many papers that are not cited presenting the effect of different metal, effect of particle size and experimental conditions for furfural hydrogenation as well as several important computational studies.

In conclusion I don't recommend the publication of the manuscript at this stage since a significant level of work has to be carried out and the findings result in a poor catalyst that actually is not stable.

However, if the authors manage to produce a stable catalyst, with more reliable level of selectivity and conversion to the desired products the impact of this work will improve considerably.

Reviewer #3 (Remarks to the Author):

The authors synthesized HCPs with solely –OH or –CH₃ groups on the polymer scaffold to tune the environment of active sites. The results revealed that HCP-OH catalysts enhance the hydrogenation rate of H-acceptor substrates containing carbonyl groups while HCP-CH₃ one promote non-H bond substrate activation. Since active sites control by optimizing the surface chemistry of the catalysts have been done by many researchers, the results seem to be promising for the selective catalysis. However, there are several concerns that the authors need to address before it could be considered for publication.

- (1) The sample name in Figure 1 b) and Figure S1 a) and b) are not correct. They are opposite.
- (2) The name of substrate and scaffolds in the caption for Figure 3 d) and e) are not correct.
- (3) It seems that oxygen functional groups especially for –OH groups are important to improve the interaction for the catalytic reaction. Authors characterized the –OH groups by some analytical techniques such as FT-IR and SS NMR. Why are oxygen contents for HCP-CH₃ and HCP-OH determined by elemental analysis omitted in Table S1? The amount of –OH groups for HCP-CH₃ and HCP-OH should be determined.
- (4) How are the catalytic results for HCP-CH₃ and HCP-OH without Ir content?
- (5) Although authors characterized the prepared samples by various techniques, some results such as XRD and porous textures toward the reactions are not discussed.

Point by point response

Reviewer #1 (Remarks to the Author):

In this manuscript, Shi and co-workers report a novel catalyst for selective hydrogenation, and tailor the activity and selectivity by augmenting functional groups on the catalyst. Experimental investigations were further appended by DFT calculations. While the idea is novel, I have some observations, issues, and reservations. Some highlights have been provided below:

Response : We appreciate finding the idea novel.

1. The authors have used the term "enzyme mimetic" on multiple occasions. Is this really appropriate? Enzyme mimetics generally implies that the reaction mechanism is same as that in the enzyme, which has not been attempted to be proved in the present case. See author's own statement: "In summary, inspired by enzyme catalysis, we designed HCPs-based catalysts with different environments consisting of precisely grafted OH and CH₃ groups around the active sites whose density can change." So I would rightly agree that the catalyst is enzyme inspired rather than enzyme mimetic.

Response : Thanks for the comment. Indeed, the origin of this paper is inspired by enzyme catalysis, mainly the weak interaction of the binding pocket with the substrate. We mimic the enzyme catalysis regarding the H bond interaction rather than the whole enzyme. We agreed the catalyst is "enzyme inspired" is more appropriate than "enzyme mimic" considering the fact that enzyme catalytic process is challenging. We have modified our expression in the revised manuscript.

2. Page 1: See the correction PMDS instead of PDMS.

Response : Thanks for pointing out the mistake. We have fixed this typo in the revised manuscript.

3. Figure 2: y-axis title could be better; sigma is not defined. This in fact is a general problem with several figure axes titles.

Response : Thanks for the suggestion. All the y-axis titles have been modified, and the definition of sigma was added.

4. Page 3: "Although the two catalysts are similar in particle size and electronic state...": I highly doubt that it would be even possible to comment upon similarity of their "electronic structures". This statement is not appropriate.

Response : Thanks for the comment. We agreed the expression is not accurate. We used XPS and CO adsorption to detect whether the Ir particles have similar electronic density. They have similar XPS peaks corresponding with Ir⁰ 4f and the CO adsorption is all located at 2044 cm⁻¹ indicating these two sites are similar. We have changed the "electronic state" to "electronic density".

5. Page 5: I would completely disagree with the computational insights provided in the manuscript. There

has been no justification on choice of the model and even why graphene was used as a representative of POPs. The DFT analysis is just not appropriate, and for the sake of correctness, I would even be willing to get rid of the computational analysis.

Response : Thanks for the constructive comment. We agree that the graphene model is insufficient to describe the structure of POPs. Unlike MOFs and COFs, the POPs are amorphous and extremely hard to model computationally. In this manuscript, however, the main goal is to demonstrate that POPs with OH groups have H-bond interaction with furfural and thus promote furfural hydrogenation, whereas POPs with CH₃ groups do not. Therefore, we mainly focus on the effect of functional groups of POPs rather than the detailed spatial configuration. The graphene models in which the terminal H is replaced by the OH group and CH₃ group, respectively, can not only be used to investigate the H-bond interaction between furfural and the OH group but also contain aromatic rings found in POPs. Besides, we also employed phenol and toluene molecules which are the monomers of HCP-OH and HCP-CH₃, respectively, to verify the role of hydrogen bond in the binding of furfural. We did find that compared with toluene, phenol binds stronger with furfural by 0.33 eV and weakens the C=O bond of furfural due to the presence of the H-bond between phenol and furfural. We have explained and highlighted these points in the revised MS and SI (Figure S22).

6. Page 5: "This is the first evidence that the OH groups play a role beyond adsorption in catalytically promoting the C=O bond hydrogenation" - This generic claim not correct. There have been several reports which reveal the role of surface functional groups beyond affecting adsorption. See for e.g. Huang et al., Matter, 2019, 1, 1656–1668; Rytter and Holmen, Topics in Catalysis, 2018, 61, 1024–1034; Xu, JACS, 2009, 16366

Response : Thanks for the insightful comment. The papers brought up regarding the OH group function has exploited CO₂ hydrogenation, CO oxidation, and Fischer–Tropsch synthesis. These reactions differ from this manuscript. To precisely express our intent, we adjusted the expression to get rid of the potential confusion. "This clearly shows that the OH groups play a role beyond adsorption in catalytically promoting the C=O bond hydrogenation. We return to this point below."

Reviewer #2 (Remarks to the Author):

The proposed manuscript titled "Selective hydrogenation via precise hydrogen bond interactions on catalytic scaffolds" reports an intriguing work devoted to demonstrating the influence of active site environment on catalytic performance. Mimicking enzyme composed of active sites and protein the authors synthesize hyper-crosslinked porous polymer modified with -OH and -CH₃ groups on the polymer scaffold to tune the interaction between reactant and active sites. This approach is quite interesting. Indeed, a fundamental understanding of the effect on active sites environments on its interaction with the reactant and on structure-performance relationship will open numerous new opportunities for the development of advanced catalysts. Similar works were reported in the recent years using zeolite and studying the confinement-induced reactivity and the molecular recognition phenomena on these catalysts (see as an overview ChemCatChem 2019,11,134– 156).

Response : We appreciate the positive evaluation.

The synthesis and characterization of the materials is well reported nevertheless, the following points should be considered by the authors for the publication in a high impact journal as Nature Comm. In particular, the part related to catalytic tests has to be improved since the current presentation leave doubt on the observed effect and on the interpretation and conclusions.

Response : Thanks for the comment. Indeed, as the reviewer comments, several works have reported an enzyme inspired catalyst by mimicking the unique 3D structure composed of an active site and a binding pocket. These studies further prove the unique value of our research, i.e., the desirability to mimic the promotion mechanism of enzymes. Despite mimicking the enzyme's 3D structure, it is even more desirable to mimic the promotion mechanism of enzymes by tuning the weak interactions between the reactant and the environment. In fact, the remarkable catalytic performance of the enzyme 3D structure is mainly achieved via the weak interaction among the substrate, the active site, and the binding pocket, for example, from hydrogen bonding and dispersion forces. Herein, inspired by the enzyme mechanism, we aimed at designing active sites with precisely controlled environments to form a hydrogen bond with substrates that act as H acceptors. Such H bond interactions enhance adsorption, potentially modify the interfacial sites to increase reaction rates, and promote selective C=O hydrogenation over C=C hydrogenation in α - β unsaturated and furanic aldehydes. This work provides a new strategy for biomimetic inspired catalysts, by taking advantage of the weak interactions between substrates and the active site environment to direct the chemistry.

- Preparation method, they used the impregnation method to produce Ir nanoparticles, not to introduce Ir nanoparticles as they write. The used it is not the best method to control the Ir particle size and could cause some variability on the studied materials due to the different hydrophilicity of the polymer support surface where applied. Small difference in nanometal particles dimension or adhesion on support can produce significant difference in catalytic activity.

Response: Thanks for the insightful comment. We apologize our writing was not so accurate. Indeed, the pre-synthesized method using PVP or other protecting agents is a powerful strategy to fabricate nanoparticles with uniform particle size. However, the impregnation method is also common to introduce nanoparticles, especially in porous structures like POPs. In this manuscript, our aim is to take the advantage of the functional groups on the skeleton and the substrate. Thus, the nanoparticles should be in the skeleton pores as much as possible. For this reason, the impregnation method is more advantageous than the pre-synthesis method. The pre-synthesized nanoparticles tend to deposit on the surface of the supports. Moreover, in order to maintain a similar particle size, we used triphenylamine with N as the binding sites to immobilize the formed nanoparticles. We also used the mixed solvent of water and alcohol to avoid surface hydrophilicity difference (see the details in experiment part).

We have changed our expression from “Ir nanoparticles were introduced via impregnation (Scheme 1; Ir-HCP-OH and Ir-HCP-CH₃) and were homogeneously distributed inside the HCPs.” to “Ir nanoparticles were introduced via metal salt impregnation and reduction method (Scheme 1; Ir-HCP-OH and Ir-HCP-CH₃) and were homogeneously distributed inside the HCPs.”

- The authors should include some schemes with the hydrogenation reaction network and possible products and better discuss in the main text of the objectives for the products they want to target.

Response : Thanks for the comment; this is a good suggestion. The scheme of the hydrogenation reaction has been added (see below) and the main target for each reaction was discussed. We emphasize that the focus of this manuscript is not to upgrade a specific platform molecule to high valued products. The furfural hydrogenation is the main model reaction we used in which selective hydrogenation of C=O over the ring is desired. Yet, we also tested other model reactions in figure 2c, 2d and figure 5 to demonstrate that these enzymes inspired catalysts expose substrate-specific catalytic performance. We added the desired product in each case. Hydrophilic catalysts form H-bonds with C=O containing substrates like furfural, absent from weakly polar substrates. Such interactions enhance adsorption, potentially modify the interfacial sites to increase reaction rates, and promote selective C=O hydrogenation over C=C hydrogenation in α,β -unsaturated and furanic aldehydes.

Scheme R1 Reaction networks for the reactions depicted in figure 2a, 2c and 2d.

- The calculation of conversion, selectivity, TOF and carbon balance should be included. When they use TOF should actually consider the number of surface-active sites calculated by other methods, not the nominal weight of metal. Moreover, possible leaching and experiments with hot filtration to verify that the reaction is not happening due to leached metal. Leached metal can be reabsorbed therefore hot filtration experiment is essential.

Response : Thanks for the comment. The conversion and selectivity were added. The yield of furfural alcohol and furfural conversion were measured by the internal standard method.

Temperature (C°)	Time (h)	Hydrogen Pressure(psi)	Conversion (%)	Yield (%)	Selectivity (%)	Carbon balance (%)
120	1	300	21.0	19.2	91.2	96.6
120	2	300	35.7	35.3	98.8	99.5
120	4	300	45.7	43.5	95.3	97.8
120	10	300	66.0	59.6	90.6	93.4
120	1	145	12.6	12.3	97.8	99.7
120	1	580	35.9	35.2	98.0	99.3

As for the TOF, it is calculated from the measured rate. For surface-active site measurement, CO pulse adsorption is commonly used. However, both catalysts showed no adsorption. The detailed reason is unknown but may be due to the porous structure of POPs, as a recent report showed that the porous POPs structure hinders the transport of gas (*Nat Catal* 2019, 2, 852–863).

A hot filter experiment was performed to exploit if leaching affects the reaction. It is observed that the reaction proceeded with time prolong (Figure R1, black line). However, when the catalyst was filtered out during the process (Figure R1 red line, the catalyst was filtered at the vertical line), the reaction totally stopped. This confirmed the catalyst is heterogeneous and little leaching occurred during the reaction. Note the solvent we used is hexane, which is non-polar and non-protic and would prevent leaching.

Figure R1 Hot filter experiment of furfural hydrogen with Ir-HCP-OH.

- When they discuss catalytic activity, they should discuss in terms of selectivity at iso-conversion or yield, this discussion is missing, so the catalytic data are not well presented.

Response : Thanks for raising this point. Due to the mild reaction conditions, no C-O bond cleavage was observed during the reaction. All the reactions maintained a high selectivity to furfural alcohol with high carbon balance (Fig. R2), so selectivity comparison at the same conversion does not provide additional information.

Figure R2. Time profile of Ir-HCP-OH demonstrating that selectivity and carbon balance are very high.

• Characterisation of the catalysts including Pd and other metals should include except TEM, HRTEM, XPS and with the TEM data should include as well the standard deviation values. This should be for fresh and used catalysts. HAADF analysis is recommended especially for a high-profile paper.

Response : Thanks for the comment. We have added TEM, HRTEM, and XPS data for Pd and Pt catalysts. Also, HAADF was added for each experiment.

Figure R3. TEM, HR-TEM and HAADF of a, Pd-HCP-CH₃, b, Pd-HCP-OH, c, Pt-HCP-CH₃, d, Pt-HCP-OH.

Figure R4. XPS spectra of a) Pd-HCP-CH₃, b) Pd-HCP-OH, c) Pt-HCP-CH₃, and d) Pt-HCP-OH.

Figure R5. Post reaction TEM images of a, Pd-HCP-CH₃, b, Pd-HCP-OH, c, Pt-HCP-CH₃, d, Pt-HCP-OH.

- Have the authors verified that they work are under kinetic regime, e.g. effect of stirring rate, amount of catalyst used and reaction temperature to calculate act. energy? What is the effect of hydrogen pressure and temperature? Have they optimised the reaction conditions? These data should be presented to demonstrate the reliability of discussed comparison.

Response: Thanks for the comment. It is important to ensure that the reactions are in the kinetic regime when we compare rates. We have made sure that rate was comparable under our conditions. All rates were measured at a conversion below 10-20%. Since the reactor we used is a stirred tank, the reaction was performed under stirring of 700 rpm to avoid diffusion effects. (Rates do not vary with the stirring speed over 500 rpm; Figure R6a below). This was also confirmed by the measured apparent activation energy of 37 kJ/mol (Figure R6d), which falls in the common range of furfuryl hydrogenation. The reaction rate increased with the hydrogen pressure and catalyst amount (Figure R6c and b).

Figure R6 a. Stir rate effect on rate with Ir-HCP-OH, b, catalyst amount effect with Ir-HCP-OH, c, hydrogen pressure effect with Ir-HCP-OH, a, Arrhenius plot of Ir-HCP-OH in furfural hydrogenation.

My main concern is the stability of the catalyst that obviously, for the results the authors present it is very low. If the catalyst is actually not recyclable, the authors should find a way to stabilise it first, otherwise there is no possible to take in account all the discussions about different environment on active site since we are looking to a transient catalytic behaviour.

Response : Thanks for this good point. The stability is important and several papers have focused on improving the stability and the deactivation mechanism. In our research, the focus is not the stability of the bio-inspired catalyst but rather on the weak interaction between the substrate and the micro-environment in promoting the reaction. We also consider the stability of the metal site when designing the catalyst by introducing N during the support synthesis (Confirmed by XPS and element analysis). A lot of papers have shown that the metal site binds N and improves the stability (ACS Catal. 2016, 6, 4, 2642–2653, Chem. Commun., 2013,49, 6623). In this manuscript, although the stability of the catalyst is not excellent, it is stable for at least two runs (Figure S24), which is sufficient to compare the activity.

In our ongoing work, we seek methods to improve it. We provide here data for review only purposes. For example, we used the 3D encapsulation method to confine the nanoparticles in the skeleton (Scheme R2). The catalyst was synthesized with sequence steps as shown in Scheme 1. The inner kernel skeleton (POF) was prepared first, and the nanoparticles were prepared by the sol-immobilization method (Figure

R7A) were deposited on it to form the kernel. Second, the kernel was enclosed by the outer shell polymer through in-situ polymerization, forming the encapsulated 3D structure. The successful encapsulation of NPs by a porous shell was verified by HR-TEM and XPS ion sputtering. Particles could be observed in the kernel (Figure R7B). However, the contrast of the NPs decreased after the kernel was encapsulated by the outer shell polymer (Figure R7C), indicating an additional layer of polymer grown out of the supported metal. Moreover, it was found that the XPS signal of Au(4f) on the surface of the encapsulate catalyst was almost missing, sharply contrasting with the kernel stated above (Figures R8a and R8b). Given the surface sensitivity of XPS, we attributed the missing signal to the kernel buried under the outer layer of POF, and the depth is at least 4-5 nm (exceeds the depth detection limit of XPS). To confirm our hypothesis, in-situ argon ion sputtering was performed to remove part of the outer shell, to obtain the depth information of elements in the layered structure sample. After argon ion sputtering on the outer surface for a certain time, a detectable signal of Au(4f) was observed (Figure R8c), confirming the missing signal was due to a thick cover of the outer polymer over AuPd nanoparticles which hindered the detection of XPS. These findings strongly support that a sandwich structure forms and NPs are encapsulated by the outer polymer. This strategy has just been reported to be effective in the preparation of ultra-stable catalysts (Nat. Mater. 2022 <https://doi.org/10.1038/s41563-022-01376-1>). We believe this strategy will improve the stability of the catalyst in our future works. We commented on this point in the revised manuscript.

Scheme R2. Schematic illustration of the synthesis path for encapsulated catalysts.

Figure R7. HR-TEM images of (A) AuPd nanoparticles; (B) AuPd NPs on inner kernel support POF-N; (C) AuPd NPs on encapsulated catalyst E-S=O, and the corresponding particle size distributions.

Figure R8. XPS spectra of (a) AuPd/POF-N, (b) E-S=O, and (c) in-situ argon ion sputtering on E-S=O.

Moreover, the authors should discuss their work and present their findings with a comparison with the current literature in terms of experimental and computational work, therefore addition of a table. There are many papers that are not cited presenting the effect of different metal, effect of particle size and experimental conditions for furfural hydrogenation as well as several important computational studies.

Response : Thanks for the comment. We have added Table S1 as suggested to compare recent furfural hydrogenation with different methods to modulate the reaction, including DFT and experiments.

Table S1. Comparison of recent works on furfural hydrogenation modulation.

Entry	Catalyst	Active site	Main products	Method to modulate catalytic performance	Method	Ref.
1	OH functional HCPs supported Ir	Ir	Furfural alcohol	Hydrogen bond interaction	Experiment + DFT	This work
2	Organic modified Pd/TiO ₂	Pd	Furfural alcohol	Surface active site modulation	Experiment	¹
3	Co-impregnation of Pd and Ir on SiO ₂	Ir,Pd	Furfural alcohol and tetrahydrofurfuryl alcohol	Metal alloy method	Experiment	²
4	Pt(111), stepped Pt(211), and Pt55 cluster	Pt	Furfural alcohol and furan	Particle size	DFT	³
5	sol-immobilisation Pd/TiO ₂	Pd	Furfural alcohol and tetrahydrofurfuryl alcohol	Particle size and site	Experiment + DFT	⁴

6	α -MoC	α -MoC	Methyl furan	Solvent	Experiment + DFT	5
7	Phosphoric acid modified Pt/Al ₂ O ₃	Pt	Methyl furan	Interfacial Metal-acid site	Experiment	6
8	Pd(111)	Pd	Furfuryl alcohol or furan	Hydrogen-coverage	DFT	7

1. Pang SH, Schoenbaum CA, Schwartz DK, Medlin JW. Effects of Thiol Modifiers on the Kinetics of Furfural Hydrogenation over Pd Catalysts. ACS Catalysis 4, 3123-3131 (2014).
2. Nakagawa Y, Takada K, Tamura M, Tomishige K. Total Hydrogenation of Furfural and 5-Hydroxymethylfurfural over Supported Pd–Ir Alloy Catalyst. ACS Catalysis 4, 2718-2726 (2014).
3. Cai Q-X, Wang J-G, Wang Y-G, Mei D. Mechanistic insights into the structure-dependent selectivity of catalytic furfural conversion on platinum catalysts. AIChE Journal 61, 3812-3824 (2015).
4. Rogers SM, et al. Tandem Site- and Size-Controlled Pd Nanoparticles for the Directed Hydrogenation of Furfural. ACS Catalysis 7, 2266-2274 (2017).
5. Deng Y, et al. Solvent Tunes the Selectivity of Hydrogenation Reaction over α -MoC Catalyst. Journal of the American Chemical Society 140, 14481-14489 (2018).
6. Zhang J, et al. Control of interfacial acid–metal catalysis with organic monolayers. Nature Catalysis 1, 148-155 (2018).
7. Wang S, Vorotnikov V, Vlachos DG. Coverage-Induced Conformational Effects on Activity and Selectivity: Hydrogenation and Decarbonylation of Furfural on Pd(111). ACS Catalysis 5, 104-112 (2015).

In conclusion I don't recommend the publication of the manuscript at this stage since a significant level of work has to be carried out and the findings result in a poor catalyst that actually is not stable. However, if the authors manage to produce a stable catalyst, with more reliable level of selectivity and conversion to the desired products the impact of this work will improve considerably.

Response: We have provided only for review synthesis and characterization data that are promising for enhanced catalyst stability and discussed in the paper ideas of how to accomplish this.

Reviewer #3 (Remarks to the Author):

The authors synthesized HCPs with solely –OH or –CH₃ groups on the polymer scaffold to tune the environment of active sites. The results revealed that HCP-OH catalysts enhance the hydrogenation rate of H-acceptor substrates containing carbonyl groups while HCP-CH₃ one promote non-H bond substrate activation. Since active sites control by optimizing the surface chemistry of the catalysts have been done by many researchers, the results seem to be promising for the selective catalysis. However, there are several concerns that the authors need to address before it could be considered for publication.

(1) The sample name in Figure 1 b) and Figure S1 a) and b) are not correct. They are opposite.

Response: Thanks for pointing out the mistake. We have fixed it in the revised manuscript.

(2) The name of substrate and scaffolds in the caption for Figure 3 d) and e) are not correct.

Response: Thanks for pointing out the mistake. We have fixed it in the revised manuscript.

(3) It seems that oxygen functional groups especially for –OH groups are important to improve the interaction for the catalytic reaction. Authors characterized the –OH groups by some analytical techniques such as FT-IR and SS NMR. Why are oxygen contents for HCP-CH₃ and HCP-OH determined by elemental analysis omitted in Table S1? The amount of –OH groups for HCP-CH₃ and HCP-OH should be determined.

Response: Thanks for the reviewer’s comments. The equipment we used does not measure the O element directly and ascribes the left weight to O element instead. Following the reviewer’s comment, we used the Boehm method to determine the OH amount (3.5 mmol/g). In addition, the O amount difference was also confirmed using the O 1s peak intensity difference. It is observed that the intensity of HCP-OH is much higher than HCP-CH₃.

Figure R9. XPS spectra of O 1s HCP-OH and HCP-CH₃.

(4) How are the catalytic results for HCP-CH₃ and HCP-OH without Ir content?

Response: Thanks for the comment. We have added the catalytic results for these two supports without Ir. Both supports showed no activity.

(5) Although authors characterized the prepared samples by various techniques, some results such as XRD and porous textures toward the reactions are not discussed.

Response: Thanks for the comment. We have added the discussion about the XRD and porous textures.

“Ir nanoparticles were introduced via impregnation and reduction method (Scheme 1; Ir-HCP-OH and Ir-HCP-CH₃) and were homogeneously distributed inside the HCPs (TEM images in Figure 1g and Figure S3). No obvious peaks of Ir were seen in XRD patterns (Figure S4) indicating Ir was in high dispersed state. A slight BET area decrease was observed upon loading Ir but the porous structure was preserved (Table S1, Figure S2).

REVIEWERS' COMMENTS

Reviewer #1 (Remarks to the Author):

The authors have addressed all the queries to my satisfaction in this version of the submission. The manuscript may now be accepted for publication.

Reviewer #2 (Remarks to the Author):

The authors have tried to answer most of the questions raised by the reviewer, however for a high impact Journal like Nat. Communications still they have not resolved important challenges.

First of all, they have not discovered an “enzyme mimetic” catalyst, this makes a huge difference, therefore the impact of the work is not high.

Secondly, the stability is a major issue and the authors have claimed they will resolve it in a future work, however for a high impact publication, the authors should solve it at this stage, otherwise the scientific impact is still low.

Metal leaching is also needed to present by adding a table in the suppl. Information. They can perform the leaching analysis of the solution after reaction or of the used catalyst.

The table of comparison with the literature and list of publications should be added in the main text.

Therefore, I still don't recommend the publication of the manuscript at this stage since a significant level of

work has to be carried out for the stability of the catalyst, and the concept of encapsulation is not novel, especially when the catalyst stability is low, and finally you can't claim that you have good stability for 2 runs, this is not important actually for a high impact work. Therefore, I still don't recommend the publication of the manuscript at this stage since more work is needed.

Reviewer #3 (Remarks to the Author):

The important factor for the catalytic reaction is OH groups. The authors characterized the amount of OH groups by Boehm titration as a bulk content together with XPS as an external surface content. The differences among the HCP-OH and HCP-CH₃ were confirmed in terms of the surface chemistry for oxygen complexes. The other issues that need to be addressed were described in the revised manuscript. It could be considered for the publication.

Point by point response

Reviewer #1 (Remarks to the Author):

The authors have addressed all the queries to my satisfaction in this version of the submission. The manuscript may now be accepted for publication.

Response : We appreciate the comments and thanks for the feedback to improve our manuscript.

Reviewer #2 (Remarks to the Author):

The authors have tried to answer most of the questions raised by the reviewer, however for a high impact Journal like Nat. Communications still they have not resolved important challenges.

Response : We appreciate the comments and are glad that our response answered most of the questions raised by the reviewer.

First of all, they have not discovered an “enzyme mimetic” catalyst, this makes a huge difference, therefore the impact of the work is not high.

Response : Thanks. We had revised our manuscript according to Reviewer #1’s comment. All the “enzyme mimetic” terms have been modified to “enzyme inspired”. These changes have been highlighted.

Secondly, the stability is a major issue and the authors have claimed they will resolve it in a future work, however for a high impact publication, the authors should solve it at this stage, otherwise the scientific impact is still low.

Response : Thanks for the comment. In our opinion, this paper’s innovative contribution and inspiration is the weak interaction between the substrate and the micro-environment in promoting the reaction. We show that the catalyst is stable enough to compare the activity and verify the idea. Indeed there are several methods to regenerate the catalyst, such as washing or re-hydrogenation, as there is little leaching during the reaction (see the following response). However, regeneration will not affect this paper's conclusion and scientific impact. Stability is important, and there are papers dedicated to improving stability. For example, in the paper *Nature Materials* 21,2022,1290-1297, they reported an encapsulation strategy to improve the stability of the catalyst. Similarly, that paper paid little attention to other catalysis aspects. Redispersion methods developed recently can be employed to combat sintering, e.g., our work in *Catal. Sci. Technol.* 12, 2920-2928 (2022) and other papers cited therein. We added a brief comment for this and the next point to the manuscript where catalyst stability is mentioned before the discussion.

Metal leaching is also needed to present by adding a table in the suppl. Information. They can perform the leaching analysis of the solution after reaction or of the used catalyst.

Response : Thanks for the comment. We have performed ICP test of Ir in the reaction solution. The Ir amount is below the detection limit of our instrument (ICP-OES 7300DV), showing little to no leaching

during the reaction. We mention this before the discussion.

The table of comparison with the literature and list of publications should be added in the main text.

Response : Thanks for the suggestion. This is a good point. The Table of comparison of various works was added in the supporting information.

Therefore, I still don't recommend the publication of the manuscript at this stage since a significant level of work has to be carried out for the stability of the catalyst, and the concept of encapsulation is not novel, especially when the catalyst stability is low, and finally you can't claim that you have good stability for 2 runs, this is not important actually for a high impact work. Therefore, I still don't recommend the publication of the manuscript at this stage since more work is needed.

Response : Thanks for the comments. We did not claim that the catalyst has good stability with two runs. The data confirms that the catalyst is stable enough to compare the activity and verify our idea. As for the stability of the catalyst, please see our comments above.

Reviewer #3 (Remarks to the Author):

The important factor for the catalytic reaction is OH groups. The authors characterized the amount of OH groups by Boehm titration as a bulk content together with XPS as an external surface content. The differences among the HCP-OH and HCP-CH₃ were confirmed in terms of the surface chemistry for oxygen complexes. The other issues that need to be addressed were described in the revised manuscript. It could be considered for the publication.

Response : We appreciate the comments and thanks for helping us improve this manuscript.